# Research on Task-Service Network Node Matching Method Based on Multi-Objective Optimization Model in Dynamic Hyper-Network Environment

**DOI:** 10.3390/mi12111427

**Published:** 2021-11-21

**Authors:** Cheng-lei Zhang, Jia-jia Liu, Hu Han, Xiao-jie Wang, Bo Yuan, Shen-le Zhuang, Kang Yang

**Affiliations:** 1School of Mechanical & Vehicle Engineering, LINYI University, Linyi 276000, China; liujiajia@lyu.edu.cn (J.-j.L.); wangxiaojie@lyu.edu.cn (X.-j.W.); 2Shandong Longli Electronic Co., Ltd., Linyi 276000, China; zhuangshenle@163.com; 3School of Mechanical and Electrical Engineering, Wuhan City Polytechnic, Wuhan 430064, China; yuanbo@whut.edu.cn; 4Anyang Institute of Technology, College of Mechanical Engineering, Anyang 455000, China; 20170028@ayit.edu.cn

**Keywords:** cloud manufacturing (CMfg), Cloud 3D printing (C3DP) device resources, complex network, manufacturing resource services, Service-oriented manufacturing (SOM), task-service

## Abstract

In order to reduce the cost of manufacturing and service for the Cloud 3D printing (C3DP) manufacturing grid, to solve the problem of resources optimization deployment for no-need preference under circumstance of cloud manufacturing, consider the interests of enterprises which need Cloud 3D printing resources and cloud platform operators, together with QoS and flexibility of both sides in the process of Cloud 3D printing resources configuration service, a task-service network node matching method based on Multi-Objective optimization model in dynamic hyper-network environment is built for resource allocation. This model represents interests of the above-mentioned two parties. In addition, the model examples are solved by modifying Mathematical algorithm of Node Matching and Evolutionary Solutions. Results prove that the model and the algorithm are feasible, effective and stable.

## 1. Introduction

In the socialization process of Advanced manufacturing systems (AMSs), the applications of the Internet of Things (IoT), big data and cloud technologies in manufacturing play increasingly important roles [1]. With regard to these concepts and techniques, proposed Service-oriented manufacturing (SOM) systems (e.g., Cloud manufacturing (CMfg) and cloud-based design and manufacturing) have attracted increasing attention from researchers [2]. In recent years, with the development of 3D printing (3DP) technologies, a variety of types of 3D printers have been developed and applied in practical production, research activities, and in bio-medicine.

In relation to manufacturing capability services, essential differences exist between Cloud 3D printing services (C3DPSs) and traditional resource services that require the participation of people [3]. Potential ways through which to allocate optimal services for random arriving tasks requires further study, especially so considering the dynamic events and uncertainty of the CMfg environment [4]. The dynamic task scheduling problems in different manufacturing systems have been studied for years [5]. Typical methods include the agent-based approach [6,7], heuristic-based approach [8], real-time process information method [9], work flow-based method and pheromone based approach [10]. The manufacturing capability service of C3DPS has similar characteristics to human behavior [11]. There are various logical relationships among their manufacturing capability services. Therefore, these relationships are not unique but rather coexistence as various types of relationships [12]. Social networks mainly refer to networks that are composed of complex connections between social individuals and individuals. According to the related theory of a social network, the manufacturing capability service in a C3DPS service platform is built into an online social network that relies on these complex relationships [13].

C3DPS involve many industries, knowledge and data scales, although they experience problems related to understanding ambiguity and non-standard terminology in the natural language description [14]. At present, the format of the exchange between information from those services and the order task information with Cloud 3D printing (C3DP) is not uniform, which is of detriment to the efficient processing of order task and service matching in the C3DPS platform [15]. To resolve the ambiguity between those services’ and order tasks’ information, complex and heterogeneous resources are required along with unified description and modeling.

Therefore, based on the cloud manufacturing paradigm, this study focuses on the task-service network node matching method based on the Multi-Objective optimization model in a dynamic hyper-network environment. According to the matching constraints (The similarity of adjacency matrix) and the objective variables (The proximity and criticality of nodes), a Multi-Objective optimization problem of network node matching is proposed, and solved using node matching by the evolutionary algorithm. Based on this constraint, it is proposed that, via a mathematical model of node matching and evolutionary solutions, the genetic algorithm can solve its multiple-objective optimization problem. Thus, the problem of C3DPS node matching in a task-service network transforms a multiple-objective optimization problem into a single optimization problem.

## 2. Related Work

As a new innovation, the Cloud 3D printing service (C3DPS),a new method based on Ubiquitous Network, Data-driven, Shared services, Cross-border integration and Mass innovation, can support the people-centered management of distributed Cloud 3D printing manufacturing resources and on-demand services based on Interconnected Personalized resources and a Flexible service. Cloud 3D printing service (C3DPS) search and matching aims to match and schedule the manufacturing tasks and service resources through the C3DPS platform with various service matching algorithms according to the user’s demand description of the target service. In the process of cloud service search and matching, service supply and demand matching should fully consider the various constraints of user demand diversity and the potential users’ preferences (including product category, processing technology, product material, layer thickness accuracy and delivery period, etc.).

At present, search and matching algorithms are mainly divided into three categories: topological matching, semantic matching and geometric matching [16]. The topological matching algorithm is a measurement of the topological relationships of entities with the same name. The semantic matching algorithm measures the semantic similarity between the reference target and the source target. The geometric matching algorithm detects the geometric similarity between the reference target and the source target [17].

Under the cloud manufacturing environment, an extensive number of 3D printing resources already exist (such as Hard/Soft resources such as 3D printing equipment resources, computing resources, software and data) as is the case for the dynamic information of order task (such as status information, quality information). However, the traditional search and matching method does not meet the user’s requirements in terms of efficiency and accuracy [18]. The supply-demand matching method of C3DPS, based on the task-service network, is a new method that combines the advantages of a complex network model, semantic similarity, and topological matching methods [19,20]. This method can realize the Multi-level matching in the matching algorithm library according to the service demand [21]. Its algorithm supports the dynamic changes of the state, quality, function and the task process of services, and also supports the matching of order tasks between network nodes.

To summarize, we propose a framework of C3DPS supply-demand matching, based on task-service network according to the operation characteristics and their resource matching requirements, which matches the nodes of Cloud 3D printing service network. Combining the theories of semantic matching and topological matching, the matching method and its implementation algorithm are performed. 

## 3. The Supply-Demand Matching Framework for C3DPS Based on Task-Service Network

In the process of Supply-demand matching and scheduling, customer needs constantly change. Specifically, for the dynamic “supply”, C3DPS are dynamic published, and the QoS between services also presents a dynamic evolution during the execution process. For the dynamic “demands”, the decomposition and execution of order tasks are dynamic, and the relationship between functional requirements and atomic tasks also dynamically evolves [22]. Therefore, a C3DPS supply-demand matching model based on the task-service network can effectively reflect the above dynamic changes in a cloud environment, achieve supply-demand matching, and optimize scheduling.

### 3.1. The Characteristics of Supply-Demand Matching of C3DPS

C3DPS Supply-demand matching refers to a virtual resource process retrieval of search and matching tools and of algorithms in the C3DPS platform, according to personalized service needs, so as to obtain the optimal solution of C3DPS resources and capabilities and meet the requirements of order tasks [23]. Compared with the service matching of the semantic web, the C3DPS search and matching tools have the following characteristics:
A wide-area distribution of C3DPS

Compared with the traditional distributed network manufacturing model, the C3DPS model has better integration, involving both soft resources and hard resources. The soft resources include software resources, data resources, computing resources, knowledge resources, human resources (such as designers, operators, experts, etc.), users’ information resources, service resources (such as consulting services, maintenance services, training services, etc.), logistics resources (such as warehouses), reverse engineering capability resources, 3D modeling capability resources, process design capability resources, simulation analysis capability resources and other resources [24]. Hard resources refer to material resources, such as transportation tools, computer tools, and other such resources.

However, the traditional semantic Web service matching method is a retrieval tool for manufacturing resources, and cannot meet the intelligent matching needs of the wide-area distribution of C3DPS. In such services, the way in which to quickly and accurately search for the services that users need is one of the key problems.

2.The dynamics of massive information

In order to ensure the optimized operation of supply–demand matching, C3DPS platform must obtain the status data information of 3D printing device resources in real-time, including the status data, the executed status data, and so on. In the cloud environment, 3D printing equipment has the characteristics of a large volume, wide area and geographical dispersion, and its attribute characteristics (load status, processing status, service quality), these 3D printing devices can dynamically changing. Therefore, there is a great amount of information in this state which has the characteristics of stochastic and dynamism [24]. The way in which to solve the efficiency problem in the supply–demand matching of mass C3DPSs, is one of the focuses of this paper.

3.The accuracy of Supply-demand matching of C3DPS

This platform has 3D printing equipment resources or cloud services for heterogeneous and massive data sets, and some resources have characteristics such as fuzzy semantic information and similar functions [25]. The supplier and the demander can perform semantic matching and transactions for various types of C3D resources or services via online search in the operating mode of the C3DPS platform. Therefore, methods by which to improve the matching accuracy presents one of the key problems that needs to be solved quickly and accurately through a search of C3D resources or services.

4.The accuracy of Multi-level matching integration

In order to enhance the accuracy of C3DPS supply–demand matching results, it is necessary to integrate multiple-levels of matching accuracy. Through the description of task service network modeling, a three-level matching process can be completed, via a dynamic description of a logical inference engine. 

In conclusion, the number of services will increase commensurately to the development of C3DPS applications. In the face of wide-area heterogeneous services, methods by which to efficiently and accurately search and match presents one of the new problems that need to be solved urgently [26]. The diversity, fuzzy semantic information and similar functions of C3DPSs aim to resolve the problems of low matching efficiency and low matching accuracy. Combined with the complex network model, semantic similarity and topology matching methods, a framework diagram of C3DPS supply–demand matching is proposed, based on the task service network.

### 3.2. The Framework of Task-Service Network Node Matching Model Based on Multi-Objective Optimization

As shown in Figure 1, C3DPS supply–demand matching is based on task-service network matching, which is the first step in supply–demand matching. As a second step of supply–demand matching, the result of network node matching acts as the input condition for the hierarchical matching of C3DPSs [27]. The specific idea is as follows: after being requested to the search engine, these network nodes are ranked by the similarity $Math_{S-T}(G_{1},G_{2})$ according to the network node matching algorithm. Once its candidate C3DPS are set, that is, $WS_{1}=\{WS_{1}^{1},WS_{1}^{2},\ldots ,WS_{1}^{n}\}$; If $Math_{S-T}(G_{1},G_{2})\geq \eta_{BaseInfo}$, the conditions will be performed to task-service network matching.

The task-service network is a topological structure of a large-scale virtual network that is composed of task service nodes and various related edge task nodes; the ontology library refers to a C3DP order task domain ontology, which mainly provides semantic support for the needs/services of a DD workshop and the maker’s transactions;The ontology library refers to a domain ontology of the C3DP order task, which mainly provides semantic support for the related description and the registration release of the C3DPS demand and the service provider;The algorithm knowledge database refers to a database that provides different types of Supply–demand matching algorithm knowledge, such as the topology matching algorithm, semantic similarity algorithm and geometric matching algorithm;The information parser is responsible for classifying and extracting the demand information in the C3DP order task network and service node information, and parsing and obtaining the basic information, I/O information (including processing accuracy, mechanical and physical properties, surface roughness, etc.) and QoS information (including time, cost, quality, fault tolerance, reliability, and comprehensive satisfaction). Here, it is provided by the requirements analysis documents and the analysis documents.The resource matcher is a function that matches C3DPS via all kinds of matching algorithms in the algorithm knowledge base [28]. This matching algorithms includes task service network node matching, based on the Multi-Objective optimization model and C3DPS hierarchical matching based on the task service network.

## 4. Mathematical Model of C3DPS Order Task Execution Evaluation Based on the AHP-TOPSIS Evaluation Model

### 4.1. The Task-Service Network Node Matching Model Based on a Multi-Objective Optimization

According to the matching constraints (The similarity of adjacency matrix) and objective variables (The proximity and criticality of nodes) of S_Net and T_Net, a Multi-Objective optimization problem of network node matching is proposed, and solved in node matching by the evolutionary algorithm [29]., A flowchart of task-service network node matching, based on a Multi-Objective optimization model, is shown in Figure 2. 

In C3DPS node matching, the network node matching method is a method that calculates the similarity of network nodes, which then converts the inter-network node matching problem, into the optimal matching problem of a weighted bipartite graph optimal matching problem, which can be solved using the genetic algorithm (GA) [30]. More specifically, the solution includes two sub-problems, namely, the calculation of task-service network node similarity and the extraction of matching nodes, as shown in Figure 3.

Where, $s(v_{i}^{1},v_{j}^{2})$ represents the similarity between nodes and in the service network. Its similarity result can be calculated by the local topology information between nodes and the relationship between “matched” nodes [31]. A diagram of C3DPS node matching based on task service network is shown in Figure 3. Figure 3a shows the corresponding relationship of S_Net and T_Net (the dotted line represents the association relationship), that is, of matched node pairs; Figure 3b,c shows that the mapping relationship is established between each service node in the task-service network.

### 4.2. The Calculation of Initial Node Similarity

Considering the local topology of $v_{i}^{1}$ and $v_{j}^{2}$ of the network $G_{1}$ and $G_{2}$, it is often not possible to obtain a good initial node similarity calculation, so the information of “matched” node pair is required [32]. 

Here, a node matching method complex network-based and calculated a node of similarity with the network $G_{1}$ and $G_{2}$, and its expression is:(1)$$s(v_{i}^{1},v_{j}^{2})=\frac{n_{M}(v_{i}^{1},v_{j}^{2})}{K_{1}\times |\Gamma (v_{i}^{1})|+K_{2}\times |\Gamma (v_{j}^{2})|-n_{M}(v_{i}^{1},v_{j}^{2})}$$
where, $n_{M}(v_{i}^{1},v_{j}^{2})$ is the number of “matched node pairs” that have connections with nodes $v_{i}^{1}$ and $v_{j}^{2}$; $\Gamma (v_{i}^{\left(k\right)})(k=1,2)$ is the set of nodes that have connections with nodes $v_{i}^{\left(k\right)}$ in the network $G_{k}$, $K_{1}$ and $K_{2}$ are the weighted value of $G_{k}$.

When the density of two networks is equal, $K_{1}=K_{2}=1$.

When the density of two networks is not equal, then:(2)$$K_{1}=\frac{2\times <k^{1}>}{<k^{1}>+<k^{2}>},\;K_{2}=\frac{2\times <k^{2}>}{<k^{1}>+<k^{2}>},$$
where, $<k^{i}>(i=1,2)$ is the average value of the task-service network $G_{k}$; $k^{1}$ is the average value of S_Net; $k^{2}$ is the average value of T_Net.

If the value of connection density between two networks are significantly different, the number of node pairs that have a higher connection density relationship with the nodes $v_{i}^{1}$ can be considered and the connection information with lower connection density can be ignored [33]. The similarity $s(v_{i}^{1},v_{j}^{2})$ between the node $v_{i}^{1}$ and $v_{j}^{2}$ is:(3)$$s(v_{i}^{1},v_{j}^{2})\approx \frac{n_{M}(v_{i}^{1},v_{j}^{2})}{\left|\Gamma (v_{i}^{1})\right|},$$
where, $n_{M}(v_{i}^{1},v_{j}^{2})$ is the number of “matched node pairs” that have connections with nodes $v_{i}^{1}$ and $v_{j}^{2}$; $\Gamma (v_{i}^{\left(k\right)})(k=1,2)$ is the set of nodes that have connections with nodes $v_{i}^{\left(k\right)}$ in the network $G_{k}$.

A schematic diagram of node similarity calculation between networks is shown in Figure 4. Two target nodes $v_{i}^{1}$ and $v_{j}^{2}$ are projected to $v_{ij}$, and the node similarity is calculated for this node pair.

### 4.3. Mathematical Model of Node Matching and Evolutionary Solutions

The optimal problem with two or two numerical objectives in a given region is called the Multiple-Objective Optimization Problem (MOP). At present, the solution of MOP is the conversion of multiple objectives into a single objective according to a utility function [34]. Therefore, an objective variable is constructed based on the node proximity and node criticality of the task-service network when the adjacency matrix similarity acts as a constraint. Based on this constraint, a mathematical model of node matching and evolutionary solutions is proposed, that the genetic algorithm can solve as a multiple-objective optimization problem. Thus, the problem of C3DPS node matching in task-service network is transformed via the multiple-objective optimization problem into a single optimization problem. 

The two objective functions of task-service network node matching are: the maximum node proximity and the minimum node criticality [35].

(1)Selecting design variables

Therefore, for obtaining the maximization of the node matching accuracy, the maximum node proximity K(M) and the criticality C(M) of the minimum node in the task-service network act as key values. Given that there is a focus on the objective variables of K(M) and C(M), the optimization variables are as follows:(4)$$f(X)=f(x_{1},x_{2})=f(C(M),K(M)),$$
where, f(C(M)) is the objective function of the node proximity in the matched nodes; f(K(M)) is the objective function of the node criticality in the matched nodes.

(2)The establishment of objective function

In order to ensure the maximization of node proximity, the objective functions of the degree of any two nodes $K_{i}$, the number of connections $K_{i}^{in}$ and the length $d(v_{i},v_{j})$ of the shortest path in the matched node are selected [36]; to ensure the minimization of the node criticality, the shortest path number $S(v_{i})$ between any pair that passes through the nodes $v_{i}$ and the shortest path number $S^{’}(v_{i})$ between any pairs that do not pass through the nodes $v_{i}$ in the matched node are selected.

(1)The maximization of node proximity

**Definition** **1** **(Node** **Proximity).***refers to the proximity of matched nodes*$v_{i}$*to the local network area.*$d(v_{i},v_{j})$*indicates that the shortest path length between any two nodes of*$v_{i}$*and* $v_{j}$*in the matched node when the matched node* $v_{i}$*is a starting point, then the proximity*$C(v_{i})$*of the node*$v_{i}$*is:*(5)$$C(v_{i})=\frac{K_{i}^{in}}{K_{i}\sum_{i\in N(C),j\in C}^{n}d(v_{i},v_{j})},i\neq j,$$*where*i∈N(C)*;*$K_{i}$*indicates the degree of proximity of the node*$v_{i}$*,*$K_{i}^{in}$*is the number of the internal nodes’ connections between node*$v_{i}$*and network C;*$d(v_{i},v_{j})$*represents the shortest path between any two nodes of*$v_{i}$*and*$v_{j}$*. The higher the value of*$C(v_{i})$*, the closer the relationship between node*$v_{i}$*and network C.**In order to reduce the calculation error, it is necessary to carry out a quantitative processing of* C(i)*, after which the quantized results of*C(i)*are:*(6)$$C(M)=\sqrt{\sum_{i=1}^{n}{\left(C(v_{i})-C(v_{i}^{\prime })\right)}^{2}},$$*where*$C(v_{i})$*is the none quantified node proximity, and*$C(v_{i})\subset [0,1)$*;*$C(v_{i}^{\prime })$*is the quantified node proximity;*C(M)*is the node proximity of*$M=\{(\nu_{1}^{1},\nu_{r_{1}}^{2}),(\nu_{2}^{1},\nu_{r_{2}}^{2}),\ldots ,(\nu_{n}^{1},\nu_{r_{n}}^{2})\}$*the nodes between*S_Net*and*T_Net.*The objective function of*C(M)*is:*(7)$$f_{1}(M)=\frac{1}{C(M)}=\frac{1}{\sqrt{\sum_{i=1}^{n}{\left(C(v_{i})-C(v_{i}^{\prime })\right)}^{2}}},$$*where*$f_{1}(M)$*is an objective function of*C(M).*The smaller the node proximity*$C(v_{i})$*and the larger the objective function*$f(C(v_{i}))$*, the more similar the structure of the two networks of*V(S_N)*and*V(T_N)*, and the higher the matching value [37]*.

(2)The minimization of node criticality

**Definition** **2** **(Node** **Neighborhood).***The neighborhood of a node*$v_{i}$*is:*(8)$$\delta_{ij}=\{v_{j}|v_{j}\in V,a_{ij}=1,j=1,2,\ldots ,n\},$$*then* $k=|\delta_{ij}|=\sum_{v_{j}\in \delta_{ij}}a_{ij}$*represents a criticality of the node*;

**Definition** **3** **(Node** **Criticality).**
*If*

k>1

*in the node*

$v_{i}$

*neighborhood*

$\delta_{ij}$

*when a degree is*

k

*,*

$S(v_{i})$

*is the shortest path number that passes through the node*

$v_{i}$

*between any node pairs, and the shortest path number without passing through the node*

$v_{i}$

*is*

$S^{\prime }(v_{i})$

*, then the Node Criticality*

$K(v_{i})$

*of node*

$v_{i}$

*is:*

(9)
$$K(v_{i})=\frac{S(v_{i})}{S(v_{i})+S^{\prime }(v_{i})},$$


*If*

k=1

*, then*

$K(v_{i})=0$

*is the criticality of node*

$v_{i}$

*. The*

$S(v_{i})$

*and*

$S^{\prime }(v_{i})$

*is calculated as follows:*
*Assuming that the key domain of the node is* T_Net*, the shortest path length between any two nodes of*T_Net*and*$v_{j}$*in the degree*k*of matched node*$v_{i}$*is:*(10)$$P(v_{s},v_{j})=\{\{v_{s},v_{j}\},\{v_{s},v_{i},v_{j}\},\{v_{s},v_{i}^{\prime },v_{j}\}|v_{i}^{\prime }\in F_{i},v\neq v_{i}^{\prime }\},$$*If there are*$w_{sj}$*shortest paths between nodes of*$v_{i}$*and*$v_{j}$*, then:*(11)$$S(v_{i})=\sum_{F_{i}}s(v_{i}),$$*where* $s(v_{i})=\left\{\begin{array}{c} \frac{1}{w_{sj}},v_{i}\in P(v_{s},v_{j})\\ 0,v_{i}\notin P(v_{s},v_{j}) \end{array}\right.$.*Assume that the shortest path length between any two nodes of* $v_{i}$*and*$v_{j}$*in the degree*k*that do not pass through nodes*$v_{i}$*is:*(12)$$S^{\prime }(v_{i})=\sum_{F_{i}}s^{\prime }(v_{i}),$$*where* $s^{\prime }(v_{i})=\left\{\begin{array}{c} 0,v_{i}\in P(v_{s},v_{j})\\ 1,v_{i}\notin P(v_{s},v_{j}) \end{array}\right.$.

Here, if the process of the node criticality $K(v_{i})$ of $v_{i}$ is quantized to [0,1), then the quantized node proximity is:(13)$$K(M)=\frac{1}{2n}\sum_{i=1}^{n}\left|K(v_{i})-K(v_{i}^{\prime })\right|$$
(14)$$K(M)=\frac{1}{2n}\sum_{i=1}^{n}\left|K(v_{i})-K(v_{i}^{\prime })\right|$$
where $K(v_{i})$ is the node $v_{i}$ criticality before quantization; $K(v_{i}^{\prime })$ is the node $v_{i}$ criticality before quantization; K(M) is the node matching criticality of the S and B networks; $v_{i}$ is the node criticality before quantization [38]; $v_{i}^{\prime }$ is the node criticality before node quantization; $M=\{(\nu_{1}^{1},\nu_{r_{1}}^{2}),(\nu_{2}^{1},\nu_{r_{2}}^{2}),\ldots ,(\nu_{n}^{1},\nu_{r_{n}}^{2})\}$ is the node matching criticality of S_Net and T_Net; The more similar the two networks are ($V(S\_N)=\{\nu_{S_{1}},\nu_{S_{2}},\ldots ,\nu_{S_{m}}\}$ and $V(T\_N)=\{\nu_{T_{1}},\nu_{T_{2}},\ldots ,\nu_{T_{n}}\}$), the smaller the K(M) and the larger the $f_{2}(M)$.

(3)The constraint condition

When a task-service network node matching model is established, its constraint is the possibility to completely match S_Net and T_Net, that is, the matching rate $\varphi (S\_Net,T\_Net)=s(v_{i}^{1},v_{j}^{2})$ of S_Net and T_Net [39]. If there is an exactly matching of S_Net and T_Net, then φ(S_Net,T_Net)=100%.

(4)The mathematical model of Multi-Objective optimization

Assuming that it is possible to match sets Ω in S_Net and T_Net, the expression of the Multi-Objective optimal network node matching model is:(15)$$\min\{f_{1}(C(M)),f_{2}(K(M))\},$$
(16)$$s.t\;\left\{\begin{array}{c} \varphi (M)=100\%\\ M\in \Omega . \end{array}\right.,$$
where $f_{1}(C(M))$ and $f_{2}(K(M)$ are optimized objective functions; C(M) and K(M) are an optimized variables; φ(M)=1 is the linear equality constraint for the variable M; M∈Ω is the linear inequality constraint for the variable M.

The two quantized objective functions of $f_{1}(C(M))$ and $f_{2}(K(M))$ are calculated to a single objective function, which is recorded as:(17)$$F(M)=w_{1}f_{1}(C(M))+w_{2}f_{2}(K(M)),$$
where $w_{1}$ is the relevant weight value of $f_{1}(C(M))$; $w_{2}$ is the relevant weight value of $f_{2}(K(M))$.

According to formula (12), the individual’s fitness value is:(18)$$fitness(M)=w_{1}f_{1}(C(M))+w_{2}f_{2}(K(M)),$$

(5)The mathematical description of a genetic algorithm

A GA algorithm is a computational model that simulates the natural evolutionary process. It has high parallelism, good universality, and global optimization capabilities. To solve the problem of the digital-analog evolution of node matching, it is necessary to design a genetic algorithm based on the actual problem, and transforms multiple objective functions and constraints into fitness functions [40]. The flowchart of task-service network node matching based on the Multi-Objective optimization model is illustrated in Figure 5.

(1)The mode of Individual coding

For $M=\{(\nu_{1}^{1},\nu_{r_{1}}^{2}),(\nu_{2}^{1},\nu_{r_{2}}^{2}),\ldots ,(\nu_{n}^{1},\nu_{r_{n}}^{2})\}$, the sequence of service node in $G_{1}$ is $\nu_{1}^{1},\nu_{2}^{1},\ldots ,\nu_{n}^{1}$, and the sequence of the service node in $G_{2}$ is $\nu_{1}^{2},\nu_{2}^{2},\ldots ,\nu_{n}^{2}$, then the matching M is determined by the sequence of $r_{1},r_{2},\ldots ,r_{n}$, that is, $M=r_{1},r_{2},\ldots ,r_{n}$ [41]. Therefore, the individual matching M is path coding by Grefenstette code.

The rule of Grefenstette cods are as follows:

(A) If the code C=(1,2,…,n) is the path between service nodes, the corresponding number $M=r_{1},r_{2},\ldots ,r_{n}$ is, respectively;

(B) Take the sequence number $g_{2}$ of the first code $r_{2}$ in C as the starting point, it deletes $r_{1}$ from C, then $C^{\left(2\right)}=C^{\left(1\right)}/\{r_{2}\}$;

(C) By analogy, according to the above rules, a string such as $g_{1},g_{2},\ldots ,g_{n}$ can be obtained, that is, of an encoded individual;

(D) Repeat the above steps until the empty set.

As can be seen from Figure 6, $C_{x}^{1}$ can be corresponded to the optional service resources of $C_{1}$. When $r_{11}$ is selected, the chromosome number is 10000⋯ and the length of this string is $C_{x}^{1}$ [42]. $C_{2}$, contains an optional service resource of $C_{x}^{2}$, and the corresponding optional service set is $\left\{r_{21},r_{22},\ldots ,r_{2K_{x}^{2}}\right\}$. For example, $M=r_{1},r_{2},\ldots ,r_{n}=45687123$ is Grefenstette code of C=(1,2,…,8). If $r_{1}=4$ and $g_{1}=4$, whereby it deletes $r_{1}$ in the C, and $C^{1}=(1,2,3,5,6,7,8)$; If $r_{2}=5$ and $g_{2}=4$, it deletes $r_{2}$ in the C, and $C^{2}=(1,2,3,6,7,8)$; and so on, for which it can be obtained for up to, 44454111, that is, the Grefenstette code of $M_{2}=12345678$.

(2)The selection and the genetic operator
(a)The selection operator


The purpose of selection is to inherit the properties of the optimized individuals (or solutions) directly from the current generation population to the next generation population. In the genetic algorithm, the fitness values of individuals is calculated and ranked by the fitness ratio method [43]. The larger the fitness value of an individual, the greater the probability of being selected. Here, the probability of selection $p_{i}$ is:(19)$$p_{i}=\frac{f_{i}}{\sum_{i=1}^{m}f_{i}},$$
where, m is the population size and $f_{i}$ is the fitness function for the i-th individual.

(b)The Crossover operation

The crossover operation is a two-point crossover from two matching individuals. In this way, a new individual is generated with a crossover operation, its probability $p_{c}$. Therefore, the Grefenstette code is used to avoid repeated serial numbers after individual crossover operations, and to prevent the occurrence of individuals with unreasonable sequences.
(20)$$P_{c}=\left\{\begin{array}{c} p_{ch}-it_{er}\times \frac{p_{ch}-p_{cl}}{it_{\max}}\begin{array}{cc} & f_{a}<f_{r} \end{array}\\ p_{ch}\begin{array}{cccc} & & & f_{r}\leq f_{a} \end{array} \end{array}\right.,$$
where $p_{ch}$ is the maximum probability of the crossover operation; $p_{ci}$ is the minimum probability of crossover operation; $it_{\max}$ is the maximum number of iterations; $it_{er}$ is the current number of iterations; $f_{a}$ is the average fitness value of the population; $f_{r}$ is the greater fitness value during the process of crossover operation.

(3)The Mutation operation

The Mutation operation is a process of re-arrangement, through which each coded individual $M=g_{1},g_{2},\ldots ,g_{n}$ can be mutated with a certain probability by means of sequence shuffling. The specific process is as follows: firstly, the random path is selected for a mutation position, and set as i; Then, $g_{i}$ can be randomly changed into one element of $\{1,2,\ldots {,n\}-\{g}_{i}\}$ which changed in accordance with the order between the bits. Therefore, the mutation operation is adopted in different stages and calculated as the mutation probability $P_{m}$.
(21)$$P_{m}=\left\{\begin{array}{c} p_{ml}+it_{er}\times \frac{p_{mh}-p_{ml}}{it_{\max}}\begin{array}{cc} & f_{a}<f_{w} \end{array}\\ p_{ml}\begin{array}{cccc} & & & f_{w}\leq f_{a} \end{array} \end{array}\right.,$$
where $p_{\mathrm{ml}}$ is the maximum mutation probability, $p_{ml}$ is the minimum mutation probability, and $f_{w}$ is the greatest fitness value in the mutation operation process.

To take the three sub-tasks of portrait 3D printing as examples, it corresponds to 4, 5, and 3 optional Cloud 3D printing services for cross mutation operations [44]. For example, the chromosome $v_{1}$ and $v_{2}$ are the random intersections at the 5-th position, then:(22)$$v_{1}=(11110000),\;v_{2}=(00001111),$$

The two generations after crossing are
(23)$$v_{1}^{\prime }=(11110111),\;v_{2}^{\prime }=(00001000),$$

For example, to assume that the 5-th gene of the chromosome v=(10001111) is mutated, because the 5-th gene of the chromosome is 1, it will turn to 0. Therefore, the chromosome $v_{1}$ will turn to $v^{\prime }=(10000111)$.

(4)The steps of Evolutionary algorithm

Here, the genetic algorithm is adopted as the evolutionary mechanism, and the specific steps of the Multi-Objective optimization mathematical model are as follows:

Step 1: Initialize the algorithm parameters. This optimization algorithm of Grefenstette coding individuals is set to the parameters of the Multi-Objective optimization mathematical model. Since the classical range of the cross rate is 0.4∼0.9, the number of network nodes is $N_{1}=N_{2}=N=100$, the maximum cross probability is $p_{ch}=0.6$, the minimum cross probability is $p_{ci}=0.4$, the cross probability is $p_{c}=0.5$, the mutation probability is $p_{m}=0.1$, and the number of termination iterations is set to T=200.

Step 2: In order to calculate the similarity of the initial node all the data of “the matched node pairs” need to be obtained. Here, the similarity of the initial node will be calculated to obtain $n_{M}(v_{i}^{1},v_{j}^{2})$ and $\Gamma (v_{i}^{\left(k\right)})$ and $K_{1}$ and $K_{2}$ of the network weights [45].

Step 3: Each initial node is sorted from high to low according to the similarity, and then the first few initial nodes with high similarity are removed. The specific threshold can be determined according to the specific situation. According to the characteristics of cultural and creative products, the network node threshold $\varphi_{Base}=0.8$ is as follows:(24)$$SN=\{n_{M}(v_{i}^{1},v_{j}^{2})|s(v_{i}^{1},v_{j}^{2})\},$$

Step 4: The genetic algorithm is optimized for all the matching sets in the network, and a Multi-Objective optimization model of the network node matching problem is proposed.

Step 5: Selection and genetic manipulation. The current iteration is $it_{er}=100$; The crossover probability is $p_{c}=0.5$; The minimum mutation probability is $p_{ml}=0.1$.

Step 6: Calculate the fitness value. In a genetic algorithm, the fitness function is usually positive. For all the individuals of the current population and the new individuals, the individual adaptive value is calculated using formula (17).

Step 7: Determine the termination condition of the algorithm. This can be determined by whether the termination condition reaches the maximum number of evolutionary generations. If is reached, the evolution will be terminated and the optimal solution is output individually; otherwise, return to step 5.

Step 8: The algorithm is terminated. their becomes the sequence set $WS_{1}=\{WS_{1}^{1},WS_{1}^{2},\ldots ,WS_{1}^{n}\}$ of C3DPS. The similarity set is $SD_{1}$.

## 5. Case Study

### 5.1. The Portrait 3D Printing Product in the Field of Cultural Creativity as an Example

Taking the portrait 3D printing product in the field of cultural creativity as an example, a process of design and manufacturing can be divided into five steps, such as portrait scanning, image design and modeling, portrait 3D printing, surface treatment and coloring, which are respectively 5, 6, 5, 4 and 6 optional services [46]. Among them, the basic information of the optional C3DPS is shown in Table 1, and the location number and time and cost information of resource transportation between cities are shown in Table 2 and Table 3.

Where n=1,2,…,5 represents the n−th resource; $f_{ij}$ represents the j−th available resource of the i−th resource, each resource has j optional candidate service resources (j>1); Y represents the cost (unit: yuan) of each available resource; T is the completion cycle of each resource (unit: hour); W is the geographic location code where the service resource is located, which corresponds to the corresponding city in Table 2. Because the geographical location of each resource is different, the cost and time of logistics must be considered. Therefore, Table 3 is a table of logistics costs and time consumption among various C3DPSs.

### 5.2. The Case Study of Node Matching in the Task-Service Network

Network node matching refers to the matching between the information required by users and the candidate service set in the C3DPS resource pool, encapsulated in the task-service network [47].

Experimental environment: the number of network nodes $N_{1}=N_{2}=N=100$, the degree $\eta_{1}=\eta_{2}=0.5$ of interaction between S_Net and T_Net.

The parameters of the genetic algorithm are: the population size of M=100; the crossover probability of $p_{c}=0.5$; the mutation probability of $p_{m}=0.05$; T=200 of termination iterations; the sampling ratio of $r=p_{r}/N=p_{r}/100$; r respectively 0.1~0.9, (note that the random sampling method has been matched pairs).

It is set the number $m_{0}=10$ of nodes in the initial network, and add these connected nodes m=10, the range of its matching accuracy is ϕ=0.18~0.78.

Therefore, the topological structure of the network is matched by the similarity between the adjacency matrices with matching nodes. As shown in Figure 7, the task-service network $G_{1}$ and $G_{2}$ are the random network.

The nodes in these networks are, respectively, $v_{i}$ and $v_{j}$, the value range i is 1~10, the probability of inter-connection with the $G_{1}$ and $G_{2}$ is 0.2 and $p_{c}=0.5$. Assuming that five pairs of nodes $\left(v_{1}^{1},v_{1}^{2}\right)$, $\left(v_{3}^{1},v_{3}^{2}\right)$, $\left(v_{7}^{1},v_{7}^{2}\right)$, $\left(v_{9}^{1},v_{9}^{2}\right)$ and $\left(v_{10}^{1},v_{10}^{2}\right)$ are matched, the similarity between node 5 in $G_{1}$ and node 5 in $G_{2}$ is:(25)$$\begin{array}{l} S(v_{5}^{1},v_{5}^{2})=\frac{n_{M}(v_{5}^{1},v_{5}^{2})}{\left|\Gamma (v_{5}^{1})|+|\Gamma (v_{5}^{2})|-n_{M}(v_{5}^{1},v_{5}^{2}\right)}\\ \;=\frac{1}{5+4-1}=\frac{1}{8} \end{array}$$
where, $n_{M}(v_{5}^{1},v_{5}^{2})$ is the number of “matched node pairs $\left(v_{5}^{1},v_{5}^{2}\right)$” that have connections with the 5-th node $v_{5}^{1}$ and 5-th node $v_{5}^{2}$ in the $G_{1}$ and $G_{2}$; and $\Gamma (v_{5}^{1})$ are node sets that have connections with $v_{5}^{1}$ and all nodes in the $G_{1}$.

Similarly, the similarity matrix, corresponding to the five node pairs between the $G_{1}$ and $G_{2}$, is calculated as:(26)$$A=\left[\begin{array}{ccccc} 0 & 0 & 0 & 0 & 0.5\\ 0 & 0 & 0 & 0 & 0\\ 0 & 0 & 0.125 & 0.2 & 0\\ 0 & 0 & 0 & 0 & 0\\ 0 & 0.2 & 0.142 & 0 & 0 \end{array}\right],$$

After this process, the matching results are $\left(v_{2}^{1},v_{2}^{2}\right)$, $\left(v_{4}^{1},v_{4}^{2}\right)$, $\left(v_{5}^{1},v_{5}^{2}\right)$, $\left(v_{6}^{1},v_{8}^{2}\right)$ and $\left(v_{8}^{1},v_{6}^{2}\right)$, where the correctly matching results are nodes 2, 4 and 5. The three candidate 3D printing resources corresponding to nodes 2, 4 and 5 are $S_{2}$, $S_{4}$ and $S_{5}$., A list of candidate 3D printing resource attributes is shown in Table 4.

According to the description of resource attributes, the basic concepts of C3DPS are extracted, such as {“Color 3D printer”, “3D model of Brahma gypsum relief decoration”, etc.}; The basic concepts of candidate C3DPS are extracted, such as the basic concepts of $S_{1}$ is {“3DP series 580 color 3D printer”, “100 mesh gypsum powder”, “3D printing”}; The specific parameters are as follows:

To set an initial threshold of network node matching $\eta_{BaseInfo}=0.8$ and a concept type weight w=(0.48,0.4,0.12).

According to the definition, it is calculated to match the services $S_{2}$,$S_{4}$, $S_{5}$ and $S_{8}$ between node information and service requirements.

So, the result of node matching is: Match(N,S)=(0.892,0.913,0.955,0.763,0.653).

If $Match(N,S_{6})<\eta_{BaseInfo}$ and $Match(N,S_{8})<\eta_{BaseInfo}$, then $S_{6}$ and $S_{8}$ are not selected.

Finally, the matching service resource set is as follows: $WS_{1}=(S_{2},S_{4},S_{5})$. As shown in Figure 8, the topology of the C3DPS network is composed of 33 nodes and 46 edges. When two nodes’ colors are the same, it indicates the C3DPSs are similar. The thickness of the edges represents the similarity between the two C3DPSs, that is, the service capability has a different relationship.

### 5.3. Results and Comparison

To illustrate the availability and efficiency of the proposed a task-service network node matching method, based on the multi-objective optimization model in a dynamic hyper-network environment, it is compared with other traditional methods, including exact matching, contains matching, implicit matching and mismatching. On the basis of a detailed analysis of each matching index, the matching index’s data sets are simulated with the traditional CMfg service modeling method and four types of similarity algorithms, based on a complex network. All the algorithms’ codes are written on the MATLAB experimental platform in a Windows 7 operating system with an Intel i3-2120 (3.30 GHz) CPU with 4G memory. In the experiments, 90% of the five data sets were selected as training sets, 10% were selected as test sets, and the number of independent experiments performed was 1000. The resulting accuracy (AUC) values of the traditional MFG service modeling method and four kinds of similarity algorithms based on complex networks are shown in Table 5:

Here, HPI indicates the hub promoted index; CN indicates common neighbors; PA indicates Priority link indicator; AA indicates Adamic-Adar; RA indicates Resource Allocation; CI indicates Combined index.

Upon comparison and analysis, the experimental results in Table 5 show that, according to the CI, the developed algorithm is more accurate than the traditional method and the existing four algorithms in five different domain networks; this algorithm further improves the accuracy of link prediction and has better prediction results for complex networks; that is, its universal applicability is better. The visualization of the NS network is shown in Figure 9.

Excluding PA, the other five indicators have the highest AUC values, which may be related to the large module degree, caused by the comparison of the traditional method and four kinds of similarity algorithms based on complex networks. Here, the running times of the algorithms are also tested. As shown in Table 6, the actual elapsed time for CI is relatively short, which shows that the efficiency of the algorithm is relatively high.

Additionally, we analyzed the efficiency index (AEI) of the algorithms under specific experimental conditions, as shown in Table 7.

Finally, the matching results of the traditional method and 4 kinds of similarity algorithms based on a complex network are given in Table 8.

## 6. Conclusions

This study focused on the task-service network node matching method based on the Multi-Objective optimization model in a dynamic hyper-network environment. In the process of C3DPS node matching, the network node matching method acts as a method that calculates the similarity of network nodes, after which the inter-network node matching problem is converted into the optimal matching problem of a weighted bipartite graph optimal matching problem, which can be solved using a genetic algorithm (GA). More specifically, the solution includes two sub-problems, namely, the calculation of task-service network node similarity and the extraction of matching nodes. According to the matching constraints (the similarity of adjacency matrix) and objective variables (the proximity and criticality of nodes) a Multi-Objective optimization problem of network node matching is proposed, and solved using node matching by the evolutionary algorithm. Finally, the model examples are solved by modifying the Mathematical algorithm of Node Matching and Evolutionary Solutions. Our results prove that the model and the algorithm are feasible, effective and stable.

## Figures and Tables

**Figure 1 micromachines-12-01427-f001:**
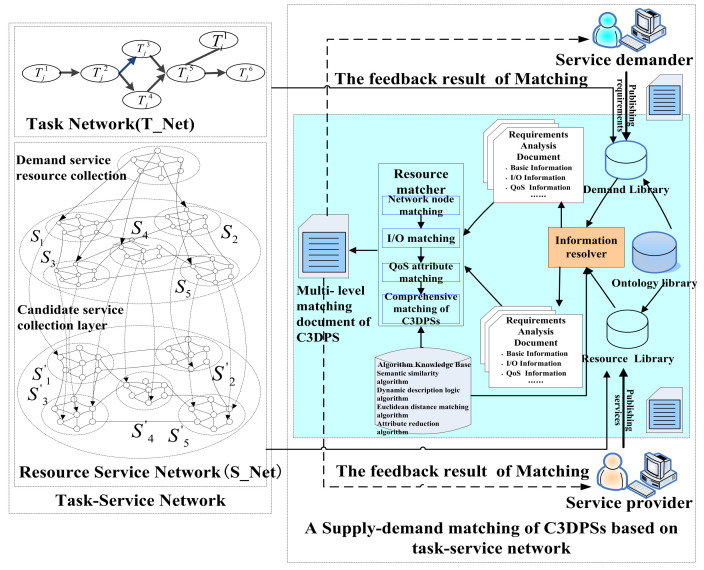
The framework of C3DPS Supply-demand matching is based on task-service network matching.

**Figure 2 micromachines-12-01427-f002:**
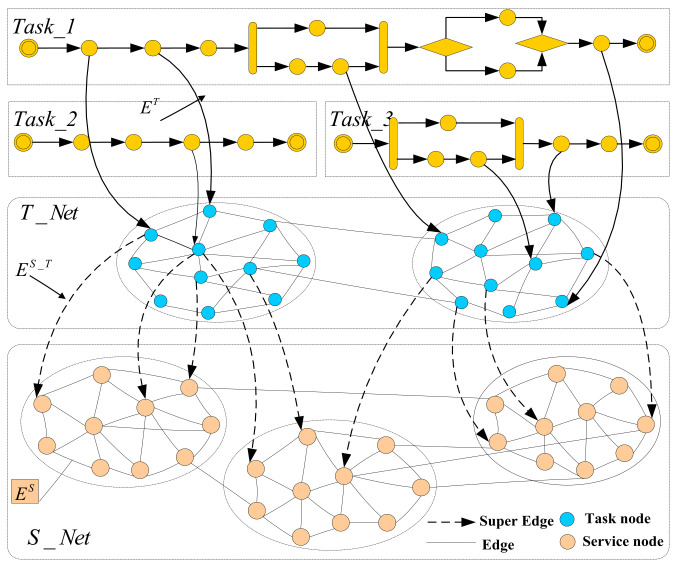
A flowchart of task-service network node matching, based on a Multi-Objective optimization model.

**Figure 3 micromachines-12-01427-f003:**
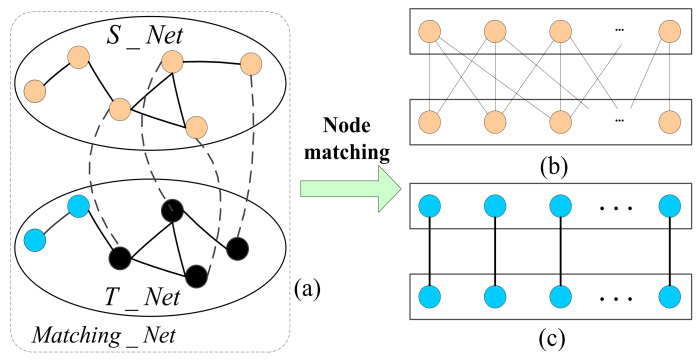
The schematic diagram of C3DPS node matching based on task-service network. (**a**) The matching_Net include S_Net and T_Net. (**b**) The mapping of C3DPSs node. (**c**) The mapping of C3DP order tasks node.

**Figure 4 micromachines-12-01427-f004:**
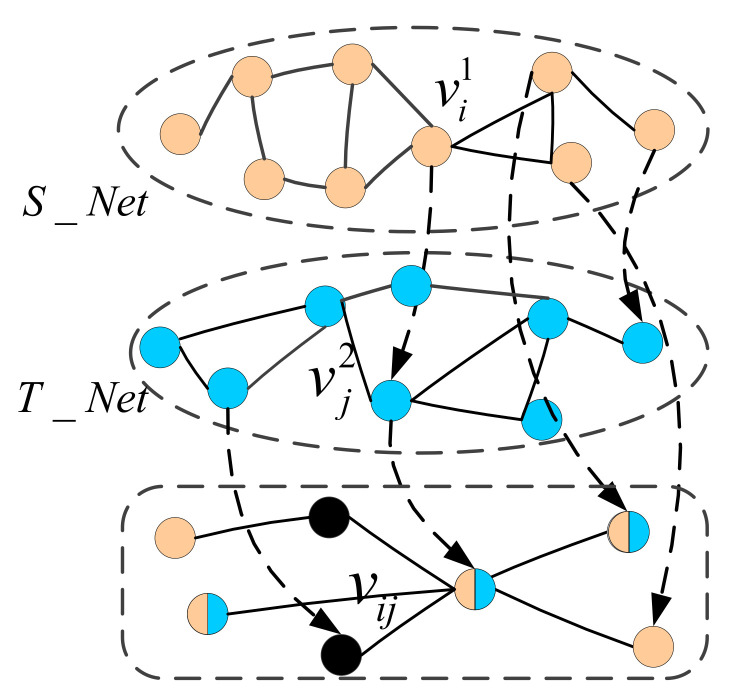
The schematic diagram of node similarity calculation between networks.

**Figure 5 micromachines-12-01427-f005:**
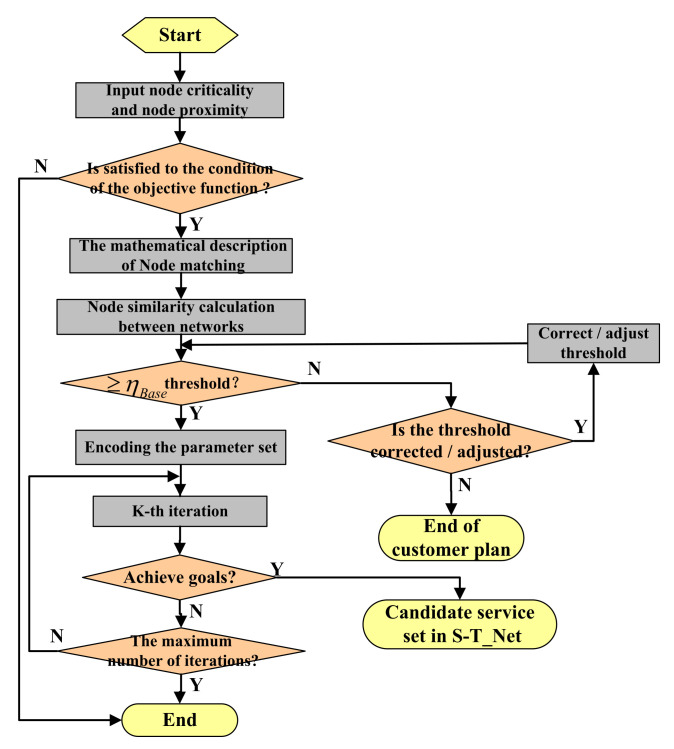
The flowchart of task-service network node matching based on the Multi-Objective optimization model.

**Figure 6 micromachines-12-01427-f006:**
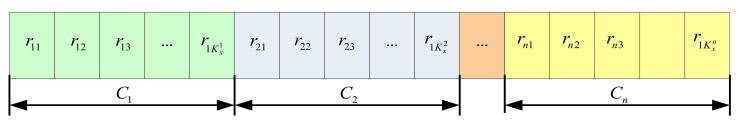
The Structure diagram of chromosome coding.

**Figure 7 micromachines-12-01427-f007:**
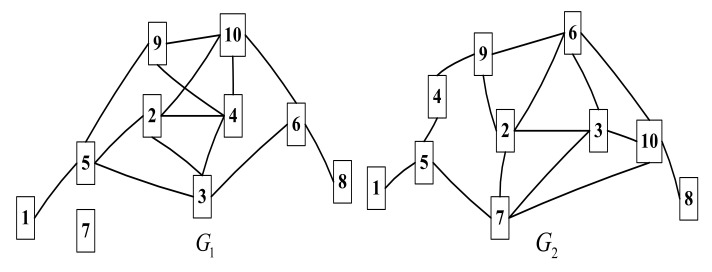
The Structure diagram of task-service network $G_{1}$
and $G_{2}$.

**Figure 8 micromachines-12-01427-f008:**
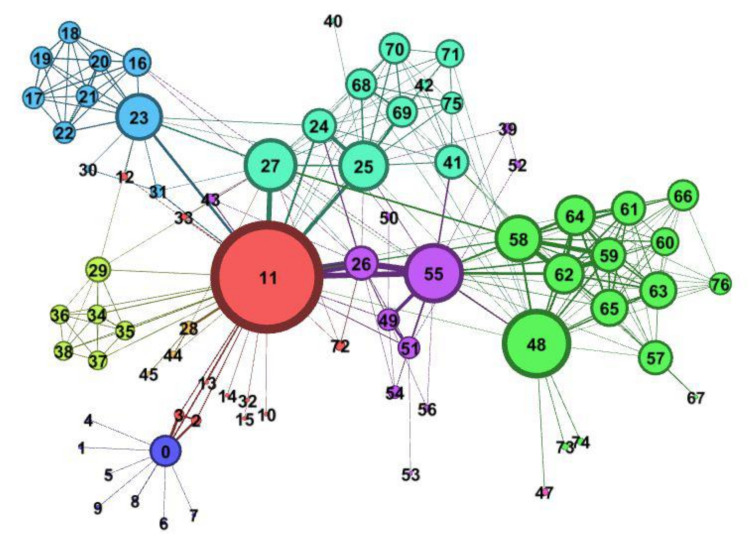
The topology of the C3DPS network.

**Figure 9 micromachines-12-01427-f009:**
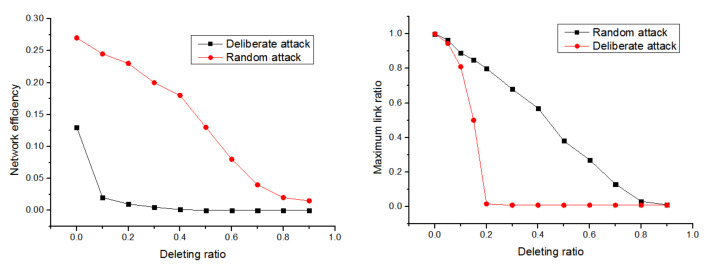
**The** visualization of the NS network.

**Table 1 micromachines-12-01427-t001:** The basic information of the optional C3DPS.

	$S_{1}$									$S_{2}$						
	$f_{11}$		$f_{12}$	$f_{13}$	$f_{14}$		$f_{15}$			$f_{21}$		$f_{22}$		$f_{23}$		$f_{24}$
Y	50		41	50	44		35			31		34		53		48
T	15		17	15	10		13			9		6		11		6
W	7		4	4	6		7			2		4		6		7
	$S_{2}$			$S_{3}$								$S_{4}$				
	$f_{25}$	$f_{26}$		$f_{31}$	$f_{32}$		$f_{33}$		$f_{34}$			$f_{41}$		$f_{42}$		$f_{43}$
Y	43	36		42	47		53		61			64		67		72
T	4	7		12	13		11		7			7		11		28
W	9	13		10	6		3		6			4		1		5
	$S_{4}$			$S_{5}$												
	$f_{44}$		$f_{45}$	$f_{51}$		$f_{52}$		$f_{53}$			$f_{54}$		$f_{55}$		$f_{56}$	
Y	74		33	25		29		27			25		30		46	
T	7		6	13		17		19			23		20		15	
W	3		8	9		12		6			8		5		4	

**Table 2 micromachines-12-01427-t002:** The basic information of the corresponding city.

Location	1	2	3	4
Address	Wuhan	Changsha	Chongqing	Zhengzhou
Location	5	6	7	8
Address	Guangzhou	Shanghai	Beijing	Nanchang

**Table 3 micromachines-12-01427-t003:** The table of logistics costs and time consumption among various C3DPSs.

	1	2	3	4	5	6	7	8
1	0/0	32/3	31/3	34/4	35/3	36/4	41/3	31/2
2	32/3	0/0	42/2	35/2	36/3	28/3	36/7	42/5
3	40/3	42/3	0/0	31/2	39/4	41/7	39/5	46/6

**Table 4 micromachines-12-01427-t004:** The list of candidate 3D printing resource attributes.

Service Name	Basic Information	I/O Information	QoS Information
$S_{2}$	Resource Name: Color 3D printer Processing material: Gypsum Service content: 3D printing	Input: 3D model of Brahma gypsum relief decoration, quantity; Output: print finished product Service scope: Wuhan Jiang’an District; Pass rate: 100%	Time: 11D Price: 2387.00 Reliability: 0.94 Environmental protection value: 93 Credibility: 4.63
$S_{4}$	Resource Name: Color 3D printer Processing material: Gypsum Service content: 3D printing	Input: 3D model of Brahma gypsum relief decoration, quantity; Output: print finished product Service scope: Wuhan Hongshan District; Pass rate: 100%	Time: 8D Price: 1986.50 Reliability: 0.92 Environmental protection value: 92 Credibility: 4.82
$S_{5}$	Resource Name: Color 3D printer Processing material: Gypsum Service content: 3D printing	Input: 3D model of Brahma gypsum relief decoration, quantity; Output: print finished product Service scope: Wuhan Hongshan District; Pass rate: 100%	Time: 14D Price: 1875.30 Reliability: 0.88 Environmental protection value: 93 Credibility: 4.78

**Table 5 micromachines-12-01427-t005:** The resulting AUC values (%) of the traditional method and 4 kinds of similarity algorithm based on a complex network.

Types		The Traditional Method				Four Kinds of Similarity Algorithms Based on a Complex Network			
		Exact Matching	Contains Matching	Implicit Matching	Mis-Matching	NS	Grid	Yeast	PB
Indexes	HPI	3.561	2.654	1.201	0.051	0.967	9.534	2.425	0.870
	CN	2.995	1.325	1.235	0.062	0.267	2.481	0.701	0.375
	PA	3.056	1.254	1.204	0.031	0.735	7.992	1.684	0.523
	AA	1.854	1.025	1.054	0.048	0.450	5.161	1.186	0.469
	RA	2.056	2.985	1.069	0.028	0.365	3.114	0.891	0.443
	CI	1.952	1.985	1.211	0.034	0.384	3.150	0.931	0.590

**Table 6 micromachines-12-01427-t006:** The operating time (s) of the traditional method and 4 kinds of similarity algorithms based on a complex network.

Types		The Traditional Method				Four Kinds of Similarity Algorithm			
		Exact	Contains	Implicit	Mis	NS	Grid	Yeast	PB
Indexes	HPI	5.6540	4.5681	3.9854	7.4501	0.5019	0.1542	0.1812	0.4106
	CN	4.8956	4.2658	3.6857	7.2822	1.8441	0.0519	0.5931	1.1370
	PA	5.6985	5.3654	4.9887	13.281	0.3361	0.0142	0.1586	0.8125
	AA	2.6542	2.3658	1.5874	8.5460	1.0824	0.0245	0.3540	0.9125
	RA	3.6521	3.0554	2.3652	16.642	1.3254	0.4089	0.4752	0.9524
	CI	6.8755	4.5697	3.6858	13.542	1.2843	0.0451	0.4525	0.7158

**Table 7 micromachines-12-01427-t007:** The efficiency (%) of the traditional method and 4 kinds of similarity algorithms based on a complex network.

Types		The Traditional Method				Four Kinds of Similarity Algorithm			
		Exact	Contains	Implicit	Mis	NS	Grid	Yeast	PB
Indexes	HPI	96.00	48.95	18.65	88.00	99.23	62.71	91.94	85.50
	CN	82.56	42.36	23.51	95.15	99.24	62.87	92.02	92.02
	PA	85.00	36.87	20.36	91.19	74.08	58.01	58.01	86.10
	AA	78.58	55.96	11.56	96.34	99.26	62.87	62.87	92.01
	RA	70.61	61.50	23.68	96.70	99.20	62.74	62.74	99.31
	CI	80.00	50.00	20.00	97.00	99.31	62.97	62.74	62.97

**Table 8 micromachines-12-01427-t008:** The matching of the traditional method and 4 kinds of similarity algorithm based on a complex network.

Types		The Traditional Method				Four Kinds of Similarity Algorithm			
		Exact	Contains	Implicit	Mis	NS	Grid	Yeast	PB
Indexes	HPI	1.000	0.700	0.400	0.000	0.923	0.838	0.919	0.855
	CN	1.000	0.700	0.400	0.000	0.924	0.728	0.920	0.927
	PA	1.000	0.700	0.400	0.000	0.756	0.881	0.880	0.861
	AA	1.000	0.700	0.400	0.000	0.993	0.887	0.873	0.920
	RA	0.800	0.500	0.200	0.000	0.992	0.774	0.874	0.993
	CI	0.800	0.500	0.200	0.000	0.993	0.897	0.861	0.630

## Data Availability

The data presented in this study are available on request from the first author.

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
