# Peer review of "Research on Task-Service Network Node Matching Method Based on Multi-Objective Optimization Model in Dynamic Hyper-Network Environment"

_micromachines, 2021, doi:10.3390/mi12111427_

Round 1

Reviewer 1 Report

The paper deals with a supply-demand matching method of Cloud 3D printing services based on task-service networks.

The paper is linear but a review of the language is required. As an example, some sentences are difficult to understand due to their length (e.g. abstract or conclusions). 

The main comments are:

  • it is not clear to the reviewer what is the novelty of the paper and how it can advance the state of the art. 
  • The introduction must better explain what C3DPS is
  • In the result section, the indicators used to compare the algorithms have to be explained
  • The conclusions have to be rewritten in order to specify the results obtained and the advancement with respect to the state of the art.

Author Response

Reviewer #1:

The paper deals with a supply-demand matching method of Cloud 3D printing services based on task-service networks.

The paper is linear but a review of the language is required. As an example, some sentences are difficult to understand due to their length (e.g. abstract or conclusions).

The main comments are:

  • it is not clear to the reviewer what is the novelty of the paper and how it can advance the state of the art.

Response: We have made correction according to the Reviewer’s comments. In the first and second part of this paper, the novelty of the paper and how it can advance the state of the art have been introduced. The modification is as follow:

Therefore, based on the cloud manufacturing paradigm, this study focuses on task-service network node matching method based on Multi-Objective optimization model in dynamic hyper-network environment. According to the matching constraints (The similarity of adjacency matrix) and objective variables (The proximity and criticality of nodes) of and , a Multi-Objective optimization problem of network node matching is proposed, and solved in node matching by the evolutionary algorithm.  Based on this constraint, it is proposed a mathematical model of node matching and evolutionary solutions that the genetic algorithm can solve its multiple-objective optimization problem. Thus, the problem of C3DPS node matching in task-service network is transformed multiple-objective optimization problem into a single optimization problem.

  • The introduction must better explain what C3DPS is.

Response: We have re-written this part according to the Reviewer’s suggestion.  C3DPS has been described in the paper.

As a new exploration, Cloud 3D printing service(C3DPS) is a new method based on Ubiquitous Network, Data-driven, Shared services, Cross-border integration and Mass innovation, can support the people-centered management of distributed cloud 3D printing manufacturing resources and on-demand services based on Interconnected Personalized resources and Flexible service.

  • In the result section, the indicators used to compare the algorithms have to be

Explained.

Response: We have made correction according to the Reviewer’s comments. In the result section, the indicator have been illustrated.

Here, HPI indicates hub promoted index; CN indicates common neighbors; PA indicates Priority link indicator; AA indicates Adamic-Adar; RA indicates Resource Allocation; CI indicates Combined index.

  • The conclusions have to be rewritten in order to specify the results obtained and the advancement with respect to the state of the art.

Response: It is really true as Reviewer suggested that We have made correction according to the Reviewer’s comments. The conclusions have been rewritten.

This study focuses on task-service network node matching method based on Multi-Objective optimization model in dynamic hyper-network environment. In the process of C3DPS node matching, the network node matching method is a method that calculates the similarity of network nodes, and then it is converted the inter-network node matching problem into the optimal matching problem of a weighted bipartite graph optimal matching problem, which can be solved by genetic algorithm (GA). Specifically, the solution includes two sub-problems, namely, the calculation of task-service network node similarity and the extraction of matching nodes. According to the matching constraints (The similarity of adjacency matrix) and objective variables (The proximity and criticality of nodes) of and , a Multi-Objective optimization problem of network node matching is proposed, and solved in node matching by the evolutionary algorithm. Finally, the model examples are sloved by modifying Mathematical algorithm of Node Matching and Evolutionary Solutions. Results prove that the model and the algorithm are feasible, effective and stable.

Reviewer 2 Report

Although the subject of the article is not directly related to my area of expertise, in my understanding the article is interesting and important contributions to science.

The conclusions seem to me unbalanced with the other sections, that is, I think the article would benefit if the authors extended the conclusions. A possible solution would be to subdivide this section into relevant subsections, such as: contributions to theory, contributions to practice, limitations and suggestions for future research.

In general, it seems to me that the article meets the necessary conditions to be published. Congratulations.

Author Response

Comments:

Although the subject of the article is not directly related to my area of expertise, in my understanding the article is interesting and important contributions to science.

The conclusions seem to me unbalanced with the other sections, that is, I think the article would benefit if the authors extended the conclusions. A possible solution would be to subdivide this section into relevant subsections, such as: contributions to theory, contributions to practice, limitations and suggestions for future research.

In general, it seems to me that the article meets the necessary conditions to be published. Congratulations.

Response: Thank you for reviewers’ comments concerning our manuscript. The conclusions seem to me unbalanced with the other sections. To solve this problem, I have rewritten this part, and modified and supplemented other chapters, and thank the judges for their comments and recognition of this paper.

Round 2

Reviewer 1 Report

The authors answer all the reviewer comments. Now the paper can be accepted.